# The Dual Function of RhoGDI2 in Immunity and Cancer

**DOI:** 10.3390/ijms24044015

**Published:** 2023-02-16

**Authors:** Mudrika Tripathi, Alain Colige, Christophe F. Deroanne

**Affiliations:** Laboratory of Connective Tissues Biology, GIGA-Cancer, University of Liège, 4000 Liège, Belgium

**Keywords:** RhoGDI2 1, RhoGDI1 2, Rho GTPases 3, cancer 4, immunity 5

## Abstract

RhoGDI2 is a guanine nucleotide dissociation inhibitor (GDI) specific for the Rho family of small GTPases. It is highly expressed in hematopoietic cells but is also present in a large array of other cell types. RhoGDI2 has been implicated in multiple human cancers and immunity regulation, where it can display a dual role. Despite its involvement in various biological processes, we still do not have a clear understanding of its mechanistic functions. This review sheds a light on the dual opposite role of RhoGDI2 in cancer, highlights its underappreciated role in immunity and proposes ways to explain its intricate regulatory functions.

## 1. Introduction

Rho GTPases are highly conserved members of the Ras superfamily, which are best known to organize the actin and microtubule cytoskeleton thereby defining the cell shape and migration. They also control a wide variety of signaling pathways that regulate crucial biological processes such as vesicle transport, cell division and gene transcription [1,2,3]. Rho GTPases cycle between an active GTP-bound form and an inactive GDP-bound form. This activity is regulated by three classes of proteins: guanine nucleotide exchange factors (GEFs) catalyze the exchange of GDP for GTP to activate the GTPase; whereas GTPase-activating proteins (GAPs) increase the intrinsic GTP hydrolysis rate of the GTPase and inactivate it; and guanine nucleotide dissociation inhibitors (GDIs) sequester the GDP-bound form of GTPases in the cytosol to prevent their activation by GEFs or ubiquitin-mediated degradation (Figure 1) [4]. Aberrant signaling of Rho GTPases and their regulators is commonly found in many human cancers and has been attributed to several mechanisms [5,6,7,8,9,10].

To this date, nearly 85 RhoGEFs and 66 RhoGAPs have been identified for nearly 20 Rho GTPase family members, wherein, in stark contrast, only three human RhoGDIs have been identified so far: RhoGDI1 (or RhoGDIα), RhoGDI2 (or RhoGDIβ or D4-GDI or Ly-GDI) and RhoGDI3 (or RhoGDIγ) [8]. All three reside exclusively in the cytoplasm wherein RhoGDI1 is ubiquitously expressed [11,12]. RhoGDI2 was initially believed to be expressed specifically in hematopoietic cells [13,14] but subsequently has also been found in various other cell types and tissues, including cancer cells [8]. RhoGDI3 is primarily expressed in the brain, lung, kidney, testis and pancreas where it targets the Golgi, and shows specificity towards RhoB and RhoG [15]. There is not much known about RhoGDI3 in cancer and immunity, therefore it will not be discussed further in this review.

RhoGDI1 and RhoGDI2 have been implicated in multiple human cancers through their involvement in cancer cell migration, invasion and metastasis and, thus, are regarded as attractive targets for cancer biology [8]. RhoGDI2 has largely remained in RhoGDI1′s shadow because of its lower abundancy and more restrained distribution. It is, however, starting to garner more attention due to discoveries hinting that RhoGDI2 may play more complex roles in multiple human cancers and many key cellular processes. This review highlights the similarities and differences between RhoGDI1 and RhoGDI2, whilst also encapsulating the multiple roles the latter has shown or has been proposed to play. Finally, we also suggest possible novel functions for RhoGDI2 and tie everything together in the context of cancer.

## 2. RhoGDI1 and RhoGDI2: Similarities and Differences

RhoGDI1 was the first RhoGDI to be discovered in rabbit intestine and bovine brain cytosol in 1989 and is widely considered to be the prototype of RhoGDIs. Subsequently, corresponding human cDNA was isolated and a RhoGDI protein was also identified in yeast [11]. Leffers et al. characterized RhoGDI2 and found that it was largely expressed in hematopoietic cells [16]. RhoGDIs interact with the GDP-bound Rho GTPases and extract Rho GTPases from the membrane to regulate them from undergoing the GDP/GTP exchange cycle. [17]. The N-terminal domain of the RhoGDIs interacts with the switch 1 and switch 2 regions of GDP-bound Rho GTPases which prevents the exchange of GDP for GTP and therefore keeps them in their inactive form [18,19], whereas the C-terminal domain also contributes towards their inhibition by extracting Rho GTPases from the membrane [17,20].

RhoGDIs may also shuttle inactive Rho GTPases towards membranes leading to their activation [17,21]. Moreover, RhoGDIs can protect its interacting Rho GTPases from proteasomal degradation [22], demonstrating that RhoGDIs are not merely inhibitors for Rho GTPases but also have a key role in their regulation and signaling. Quite expectedly in view of these functions, both the RhoGDIs are involved in the regulation of multiple biological processes such as actin cytoskeletal organization, cell migration and immune response [23,24,25,26]. As mentioned previously, they are also implicated in many human cancers where they can either be upregulated or downregulated (Table 1). The correlation between RhoGDIs and prognoses in multiple human cancer will be discussed in a later section.

Although their extreme N-terminal domain (25 and 22 amino acids for RhoGDI 1 and 2, respectively) are completely divergent, RhoGDI 1 and 2 show 73.6% identity for the remaining C-terminal sequence (Figure 2). RhoGDI1 and RhoGDI2 interact with and form complexes with the classical Rho GTPases, i.e., RhoA, RhoC, Rac1, Rac2, Rac3, RhoG and Cdc42 [19,54,55,56]. However, the interaction potency of RhoGDI2 with Cdc42 is 10–20 folds lower than that of RhoGDI1. Platko et al. observed that a single residue (Ile 177 in RhoGDI1/Asn 174 in RhoGDI2) is responsible for this difference in their affinity for Cdc42 [57].

Several other proteins that are not part of the Rho GTPase family have been found to interact with RhoGDI 1 or 2 or both, mainly through high throughput experiments. Upon examining Uniprot (https://www.uniprot.org (accessed on 23 January 2023)) and Biogrid (https://thebiogrid.org (accessed on 23 January 2023)) databases, the interactors of both RhoGDIs can be extrapolated. Both RhoGDI 1 and 2 have been found to interact with ubiquitin-fold modifier 1 (UFM1), small ubiquitin-like modifier 4 (SUMO4), U2 small nuclear RNA auxiliary factor 2 (U2AF2) and DEAD (Asp-Glu-Ala-Asp) box polypeptide 58 (DDX58). However, RhoGDI1 interacts with Cullin3, whereas RhoGDI2 does not. RhoGDI1 also interacts with EWS RNA-binding protein 1, ezrin, moesin and radixin. On the other hand, RhoGDI2 interacts with RhoGEF Vav1, whereas RhoGDI1 is unable to do so. RhoGDI2 also interacts with acyl-CoA thioesterase 7, B cell CLL/lymphoma 6 and cadherin1. Perhaps the differences in functions of both RhoGDIs may be attributed to their interactions with different proteins that do not belong to the Rho GTPase family.

Mouse models were used to identify the respective functions of RhoGDI 1 and 2. Yin et al. generated RhoGDI2-null mice to explore its functions in lymphocytes. They observed that there were no abnormalities in lymphoid development and immune responses. However, in vitro cultivation of B and T cells from these mice showed de-regulated interactions and other impaired phenotypes. They inferred that RhoGDI2 regulates Rho GTPases in lymphocyte survival and responsiveness, wherein the absence of RhoGDI2 can be compensated in vivo by other Rho GTPase regulatory proteins [26]. It was later shown that RhoGDI1-null mice display abnormalities in the kidneys and reproductive system in adulthood and that levels of RhoGDI2 expression did not change in WT and RhoGDI1-null mice [58]. Double knockouts of RhoGDI 1 and 2 were then generated in order to get a better insight into their specific and shared functions. These mice are characterized by aberrant homeostasis of lymphocytes and an increased eosinophil population. T cells derived from the mice display defective in vitro proliferation and development and lower levels of CD3 expression. These results show that RhoGDI 1 and 2 share similar functions and can partly substitute for each other in lymphocytic migration and development [59].

## 3. Regulatory Functions of RhoGDI2

RhoGDI2 is implicated in multiple biological processes in the human body either due to their regulation of Rho GTPases or independently (Figure 3).

### 3.1. Actin Cytoskeletal Organization

The three best characterized Rho GTPases are RhoA, Rac1 and Cdc42. They transduce signals in response to a chemical or mechanical stimuli to regulate different signaling pathways [60]. These Rho GTPases have been shown to modulate actin cytoskeleton organization, thereby defining cell shape and movement. RhoGDI2 sequesters Rho GTPases in the cytosol and may cause a rounding up of cells in various cell lines upon overexpression [16].

CRIF-1 (or CR6-interacting factor-1) is involved in mitochondrial functions and the regulation of cell growth. In human umbilical vein endothelial cells (HUVECs), RhoGDI2 is upregulated upon CRIF-1 silencing, which results in reduced cell migration possibly through the regulation of the activity of Rho GTPases [25]. In addition, RhoGDI2 has been shown to negatively regulate trophoblast migration via the inhibition of Rac1 activity. Trophoblasts need to proliferate and migrate to ensure a successful pregnancy [61].

### 3.2. Immune Response

#### 3.2.1. Innate Immune Response

The innate immune system is the body’s first line of a quick and non-specific mechanism of defending itself against foreign organisms entering the body [62].

RhoGDI2 negatively regulates Fcγ receptor (FcγR)-mediated phagocytosis in Jurkat T cells by preventing the localization of Rac1 to the membrane and thereby inactivating it. Phagocytosis, or the process of engulfing a foreign particle, is driven by a finely controlled rearrangement of the actin cytoskeleton. It is essential to regulate the actin cytoskeleton as phagosome and multicomponent signaling pathways transduce signals from phagocytic receptors to the cytoskeleton. FcγR is one such receptor that clusters on monocytes and macrophages to activate multiple signaling cascades to initiate phagocytosis; its function has also been implicated in the pathophysiology of autoimmune diseases and in mediating cytotoxic effects of monoclonal anti-tumour antibodies [63,64].

Though not related to its function, it is worth mentioning that RhoGDI2 was recently identified as an extremely sensitive fecal biomarker to quantify gut inflammation in patients with inflammatory bowel disease (IBD) to estimate IBD severity [65]. Severe IBD can lead to gastrointestinal cancers as well as cardiovascular diseases; immunological disorders and inflammatory biomarkers can be exploited for information about the immune system of an individual, even if they aren’t directly involved in inflammation [66].

Using mRNA microarray analysis, RhoGDI2 was found to be differentially expressed in chronic chagasic cardiomyopathy (CCC), a myocardial disease characterized by severe cardiac inflammation leading to heart failure. RhoGDI2 was identified as one of the top genes, along with Rac2, from a module that is enriched in the pathway of natural killer (NK) cell-mediated cytotoxicity [67]. This suggests that RhoGDI2 may be involved in CCC pathogenesis via NK cell-mediated cytotoxicity, possibly via Rac2 regulation [68]. Despite the fact that NK cells belong to the family of T and B cell lymphocytes, the level of RhoGDI2 protein expression in NK cells has not yet been documented [69].

The association of RhoGDI2 and Rac2 was also reported in a signaling pathway that is implicated in the pathogenesis of oral lichen planus (OLP), a chronic inflammatory and immune-mediated disease affecting skin, nail, hair and mucous membranes. Caspase-1, an inflammatory caspase, is upregulated in OLP tissues wherein Rac2 and RhoGDI2 were identified as its potential interactors. Moreover, it was also observed that the gene and protein expression levels of Rac2 and RhoGDI2 are upregulated in OLP tissues, where they positively correlate with Caspase-1 leading to immune deregulation and inflammation in OLP [70].

#### 3.2.2. Adaptive Immune Response

The highly specific and relatively slow adaptive immune system, made up of B and T cell lymphocytes and antibodies, takes over when the innate immune system is unable to destroy the germs [62].

Though also not a function of RhoGDI2, it was recently reported through a large nationwide cohort study that antibodies against RhoGDI2 can be used as a diagnostic biomarker for long-term kidney graft loss and fibrosis in patients with kidney transplants from deceased donors. Further studies showed that RhoGDI2 autoantibodies were associated with chronic antibody-mediated rejection and inferior graft survival [20,71,72,73,74]. It would be interesting to explore if the autoantibodies against RhoGDI2 can be used as a diagnostic biomarker in other organ transplants.

#### 3.2.3. Phosphorylated RhoGDI2

RhoGDI2 phosphorylated at unspecific site(s) was immunoprecipitated from Jurkat T cells stimulated by phorbol myristyl acetate (PMA). However, it could not be immunoprecipitated from resting T cells, suggesting that phosphorylation of RhoGDI2 could have potential consequences for lymphocyte activation [14]. Phosphorylated RhoGDI2 was also immunoprecipitated from monocyte U937 immune cells after PMA treatment. Further experiments showed that RhoGDI2 was phosphorylated at unspecified tyrosine residue(s) in anti-CD3 monoclonal antibody (mAb)-activated Jurkat and Raji T cells. It was also observed that RhoGDI1 remained unphosphorylated in PMA-induced U937 cells and anti-CD3mAb-activated T cells. This is of interest, as PMA induces the differentiation of U937 cells from non-adherent myelomonocytic cells to adherent macrophage-like cells and anti-CD3 mAbs are a well-recognized protocol for T cell activation.

It was also reported that upon T cell activation by anti-CD3 mAbs, RhoGDI2, but not RhoGDI1, interacts with Vav1, a RhoGEF that is also expressed abundantly in immune cells. Both Vav1 and RhoGDI2 then co-localize at the periphery of the immune synapse, the interface between the T cell and an antigen-presenting cell [75,76]. Theoretically, RhoGDI2 and Vav1 have opposing functions, so their interaction is intriguing. This could suggest that RhoGDI2 might be involved in the regulation of hematopoietic-specific Rho GTPases via different mechanisms leading to their inhibition. In Jurkat T cells, knocking down RhoGDI2 reduces their adhesive and migratory ability by decreasing the activities of Rac1 and Cdc42. It has also been recently discovered that RhoGDI2 can be phosphorylated at Tyr24, Ser31 and Tyr153 (by Src, c-Abl and Syk), which would play a role in Rac1 activation by RhoGDI2. Phosphorylation of RhoGDI2 at Y24/153 increased its ability to bind to Vav1, which could explain the promoting role of RhoGDI2 in T cell adhesion and migration, while the phosphorylation at S31 is required for the opening of RhoGDI2 and the subsequent release of its interacting Rho GTPases [24,76]. This is in contrast with the anti-migratory effect of RhoGDI2 in multiple other cell types [62], bolstering the fact that perhaps RhoGDI2 participates in different regulatory pathways in T lymphocytes. This is of additional importance as RhoGDI2 was initially observed in B and T lymphocytes where it is abundantly expressed [16], suggesting that RhoGDI2 is the predominant RhoGDI in hematopoietic cells and that it plays a dual role in the immune response.

In human platelets, RhoGDI1 has a homogenous distribution whereas RhoGDI2 is distributed unevenly or in a polarized manner within the same cell. Downregulating RhoGDI2 in human platelets inhibited their spreading, which was not observed upon RhoGDI1 silencing. It was also noticed that RhoGDI2, and not RhoGDI1, was phosphorylated at PKC substrate motifs upon platelet activation and co-localized with PKC in adherent platelets. This study suggests that RhoGDI2 may also play a role in platelet function by regulating Rho GTPases activity and affect the hemostasis [77]. This could mean that despite RhoGDI1 being the preferred binding partner for Rac1 and Cdc42 in most cell types, these key Rho GTPases preferentially bind to RhoGDI2 in human platelets.

### 3.3. Apoptosis

Apoptosis driven by caspase-3 can be induced by anti-Fas antibody, etoposide, taxol, daunorubicin, overexpression of PMA and staurosporine [78,79,80]. RhoGDI2, but not RhoGDI1, is cleaved at Asp19 by caspase-3 during apoptosis in Jurkat T cells, ML-1 and HL-60 human leukemic cell lines and macrophages. The cleaved form of RhoGDI2 (deprived of its N-terminal extremity) is then translocated into the nucleus suggesting a pro-apoptotic role of RhoGDI2. An identical caspase-3-dependent cleavage is also observed in spontaneous and TNF-α induced apoptosis in polymorphonuclear neutrophils (PMN) and in camptothecin-induced apoptotic human promyelocytic cells. This is of particular importance as apoptosis is a critical step when PMN migrates into affected tissues and interacts with extracellular matrix (ECM) proteins to ameliorate inflammation [81]. Cleavage of RhoGDI2 by caspase-3 is now considered a hallmark of the apoptosis [20,80].

### 3.4. HIV-1 Replication

Rho GTPases have been reported to regulate the replication of human immunodeficiency virus type 1 (HIV-1), the causative agent of acquired immunodeficiency syndrome (AIDS), by promoting its entry and its infection into T cells [82]. It was noted, however, that RhoGDI2 negatively regulated the rate of HIV-1 replication in infected MT-4 T cells by reducing the F-actin content and the activity of both Rac1 and RhoA at the early phase of the viral life cycle. Env are glycoproteins anchored in the viral membrane essential for the viral entry into cells, and the activation of RhoA mediates the reorganization of the cytoskeleton to facilitate this entry [83,84]. Upon further analysis, it was observed that expression of recombinant RhoGDI2 affected Env-mediated processes, possibly via receptor clustering and virus-cell membrane fusion.

### 3.5. Vascular Remodeling

Vascular remodeling is involved in the pathogenesis of many life-threatening cardiovascular diseases, transplantation and chronic rejection [85,86]. Vascular smooth muscle cells (VSMCs) undergo a transition from contractile to proliferative phenotype for successful vascular development and remodeling. Upon binding of angiotensin (Ang) II, a peptide hormone, to its receptor angiotensin II receptor type I, RhoA becomes activated and further mediates migration of VSMCs. A recent study also reported that Ang II induces proteasomal degradation of RhoGDI 1 and 2 by upregulating ubiquitination, which leads to reduced proliferation in VSMCs [87].

Consistent to this, the expressions of RhoGDI1 and RhoGDI2 are significantly increased by TGFβ1 in human aortic adventitial fibroblasts, wherein TGFβ1 drives the differentiation of vascular myofibroblasts from fibroblasts, and positively correlates with vascular remodeling [88]. Upon further investigation, it was discovered that TGFβ1 promotes the interactions between RhoGDI2 and Rac1 or Cdc42 via the Smad signaling pathway, with no effect on the interactions between RhoGDI1 and Rac1/Cdc42. This suggested that RhoGDI2, and not RhoGDI1, participates in TGFβ1 induced myofibroblast differentiation [89].

## 4. Regulatory Functions of RhoGDI2 in Cancer

RhoGDI2 is either upregulated or downregulated in many cancers depending on its type and stage. More interestingly, its level of expression can be either meaningless or strongly correlated with a good or poor prognosis. Figure 4 reports different cancer types for which studies identified RhoGDI2 as having pro- or anti-cancer tumour effects. This section reviews recent findings on RhoGDI2 in cancer progression and highlights its dual functions.

### 4.1. Ovarian Cancer

While RhoGDI1 is uniformly expressed in multiple ovarian (OV) cancer cells, RhoGDI2 protein expression is either upregulated or downregulated depending on the OV cancer cell line. RhoGDI2 has also been shown to act as a tumour and metastasis suppressor in OV cancer by enhancing Rac1 activity to activate p38 and JNK/MAPK cascades [38]. This was corroborated by a recent study that observed that Adenosine (Ado), a purine nucleotide exerting anti-tumour activity in multiple human cancers, enhances RhoGDI2 expression in A2780 OV cancer cells and A2780 subcutaneous xenografts in nude mice. They also observed that Ado inhibited tumour growth in the mice in a RhoGDI2-dependent manner [39].

### 4.2. Breast Cancer

Similar to OV cancer, RhoGDI2 protein expression was shown to be either upregulated or downregulated depending on the breast cancer (BRCA) cell line, while RhoGDI1 is similarly upregulated in all BRCA cell lines [30,31]. Eventually, a stage-dependent biphasic pattern of RhoGDI2 expression has been evidenced in breast cancer tissues with a marked increase from normal to hyperplasia, followed by a decrease from in situ to invasive lesions. An inverse correlation between RhoGDI2 expression and lymph node metastasis was observed, implying that RhoGDI2 might act as a tumour promoter but metastasis suppressor in BRCA [32]. However, it is important to remember that RhoGDI2 is not only expressed by cancer cells but also by cells of the tumour micro-environment, such as immune cells. Therefore, modifications in RhoGDI2 expression measured in tumour samples can be influenced also by differences related to immune cell populations infiltrating the tissue.

Another study further demonstrated that Rictor/mTOR pathway, which is upstream of Rho GTPases signaling, downregulates RhoGDI2 expression and promotes Rac1 activity, thereby stimulating cell invasion and metastasis in BRCA cells. They confirmed that lower RhoGDI2 levels were associated with poor prognosis in the human epidermal growth factor receptor 2 (HER2)^+^ BRCA [33]. However, the data obtained from public databases including all BRCA subtypes and analyzed with the “Kaplan-Meier Plotter” (an online tool available for exploring the correlation between the expression of any gene and survival of patients with cancer (https://kmplot.com (accessed on 25 January 2023)) [90], show that a high RNA expression level of RhoGDI2 is associated with a lower probability of overall survival (Figure 5).

### 4.3. Bladder Cancer

RhoGDI2 has largely been reported to be associated with a good prognosis in bladder cancer (BLCA) patients, where it represses metastasis by inhibiting Rac1 activity. However, it does not affect the primary tumour growth [37].

Despite being accepted as a metastasis suppressor, recent reports from Huang lab unanticipatively find RhoGDI2 levels to be consistently elevated in most human and mouse BLCA tissues. They have identified three different pathways—RhoGDI2/miR-200c/JNK2/Sp1/MMP2 pathway, XIAP/Erk/nucleolin/RhoGDI2 and, lastly, miR-145/Sp1/USP8/AUF1/RhoGDI2 pathway—through which they show that RhoGDI2 promotes BLCA invasion in vitro and lung metastasis in vivo. They observe that RhoGDI2 acts as a tumour and metastasis promoter, despite being widely demonstrated in the past as a metastasis suppressor, with no effect on the tumour [91,92,93]. Further investigations are required to better define the exact function of RhoGDI2 in BLCA.

### 4.4. Osteosarcoma

WSB1 (hypoxia-driven-WD repeat and SOCS box containing 1) is upregulated in many human cancers and acts as a suppressor of cytokine signaling. In osteosarcoma (OS), RhoGDI2 (but not RhoGDI1) is degraded by WSB1. In U2OS and MG63 human osteosarcoma cell lines, it results in the activation of Rac1 which increases the amount of F-actin, promotes the formation of lamellipodia and leads to enhanced cell motility and migration ability. WSB1 was also shown to drive the metastatic potential of OS in vitro and in the BALB/c (nu/nu) mouse model. Of most interest in this context is that RhoGDI2 overexpression in OS cells and mice is able to reverse the spreading of the WSB1-induced metastasis [50,51], demonstrating its direct participation in OS progression.

### 4.5. Leukemias

RhoGDI2 was found to be mutated at two positions (V68L and V69A) in KM3 and Reh human acute myeloid leukemia (AML) cell lines. Neither wild-type (wt) nor mutated (mt) RhoGDI2 overexpression altered cell proliferation. However, overexpressing mtRhoGDI2 promoted cell adhesiveness and invasiveness in vitro. These induced phenotypes were reversed by the overexpressing of recombinant wtRhoGDI2, which probably illustrates a competition between wtRhoGDI2 and mtRhoGDI2 for a common interacting partner that regulates AML progression [52].

CXCL12 is a chemokine that binds primarily to CXC receptor 4 (CXCR4) to induce several signaling pathways related to chemotaxis, cell survival and proliferation and gene transcription. RhoGDI2 is expressed quite highly in all acute lymphoblastic leukemia (ALL) cell lines, including Jurkat T-ALL cells. It was found that downregulation of RhoGDI2 increases CXCL12-driven T-ALL migration. RhoGDI2 mutants were created to mimic phosphorylation induced by PMA on Y24 and Y153 (discussed in previous section). These phosphomimetic mutants were seen to rescue the inhibiting phenotype induced by wtRhoGDI2 on CXCL12-mediated ALL migration. Additionally, phosphorylation of RhoGDI2 on Y24 or Y153 reduce its affinity for RhoA or RhoC, leading to their increased activity. Further investigations showed that the phosphorylation of RhoGDI2 was due to non-receptor protein tyrosine kinases Src and ABL1 in response to CXCR4 stimulation by CXCL12 in T-ALL [24,53,76,94].

### 4.6. Hodgkin’s Lymphoma

RhoGDI2 gene expression is selectively downregulated in Hodgkin’s lymphoma when compared with B cell non–Hodgkin’s lymphoma (B-NHL) cells [40]. Due to RhoGDI2′s abundance in hematopoietic cells and its possible role in apoptosis (explained in the previous section), a study attempted to evaluate the functional relevance of RhoGDI2 in apoptosis. Hodgkin L428 cells, which do not express endogenous RhoGDI2, were modified to conditionally express recombinant RhoGDI2. Its induction led to only moderate levels of apoptosis, questioning its importance in this model [41].

### 4.7. Pancreatic Cancer

RhoGDI2 was found to be significantly upregulated in tumour samples from patients with pancreatic adenocarcinoma (PDAC) wherein its expression positively correlated with tumour size, differentiation, clinical stage, lymph node metastasis, vascular invasion and reduced life expectancy (see Figure 5). Silencing of RhoGDI2 was shown to decrease the invasiveness of PDAC cells in vitro, and to reduce the expression of matrix metalloproteinase (MMP) 2, possibly through regulation of Rac1 activity [44]. It suggests that RhoGDI2, or the pathway it regulates, could be an interesting target in that particular highly aggressive cancer for which the current therapeutic options are of limited long-term efficacy. IFN-γ is a cytokine predominantly produced by natural killer cells and has anti-tumourigenic effects [45]. It was observed that IFN-γ suppresses the expression of RhoGDI2, which reduces Rac1 activity and CXCL8 expression. Moreover, overexpression of RhoGDI2 in PDAC cells in vitro enhanced their proliferation, migration, invasiveness and apoptotic resistance to Gemcitabine (a chemotherapeutic drug). It does so by increasing the expression of vimentin and Snail, a master regulator of epithelial-to-mesenchymal transition (EMT), and by decreasing the expression of the epithelial marker E-cadherin. As an additional validation of the involved mechanisms, the downregulation of Snail expression inverted the phenotypes seen by overexpressing RhoGDI2 [46].

### 4.8. Colorectal Cancer

RhoGDI2 gene and protein expressions are upregulated in the highly invasive and metastatic colorectal cancer (CRC) cell lines as compared to less invasive cell lines. It is also upregulated in CRC tissues wherein its expression negatively correlates with patient survival. Overexpression of RhoGDI2 enhances the proliferation, migration and invasive capacities of CRC cells partly via the activation of the PI3K/Akt pathway [28]. Although not explored further, the activation of PI3K/Akt pathway by RhoGDI2 overexpression in CRC could occur via its regulation of Rho GTPases.

### 4.9. Hepatocellular Carcinoma

RhoGDI2 has been shown to be upregulated in hepatocellular carcinoma (HCC) cells where its overexpression increased their rates of proliferation and invasion, via activating the PI3K/Akt pathway and increasing the levels of MMP2 and MMP9 (consistent to PDAC). MMPs are ECM-degrading enzymes involved in tumour invasion and metastasis. They are also intimately involved in the regulation of the activities of cytokines and cytokine receptors [95]. No further research was conducted to delve into the relation between RhoGDI2 and PI3K/Akt pathway in HCC [35].

A recent report claims that synergistic effects of traditional Chinese medicines *Euphorbia Pekinesis* and *Glycyrrhiza glabra* against HCC ascites in mice models is due to the downregulation of the expression of RhoGDI2 and Frk (Fyn-related kinase), a member of the Src non-receptor tyrosine kinase family. Frk and RhoGDI2 are both involved in the pathway of vasopressin-regulated water re-absorption, which has been implicated in fluid retention in cirrhosis and seems to be a common therapeutic target pathway of various anti-ascites drugs [96].

### 4.10. Gastric Cancer

RhoGDI2 shows significantly elevated expression levels in gastric cancer (GC) samples as compared to normal and para-cancerous tissues. In addition, knocking down RhoGDI2 expression decreases the migration and invasion of GC cells, enhances their sensitivity towards chemotherapeutic drugs and reduces tumour growth in mice models. RhoGDI2 acts as a scaffold protein to promote the binding of Rac1 to FilaminA, an actin binding protein. FilaminA further promotes Rac1 activation via binding to Trio, a Rac1 specific RhoGEF, and thereby enhances the invasive abilities of GC cells. Both RhoGDI2 and FilaminA are indicated to be associated with poor prognosis of GC patients. Whether RhoGDI1 can also associate with FilaminA is not known. In addition to promote Rac1 binding to FilaminA, the RhoGDI2-Rac1 axis is also involved in another signaling pathway in GC since suppressing the expression of RhoGDI2 repressed the Rac1/Pak1/LIMK1 axis, which is involved in EMT, cell migration and invasion in multiple human cancers [42,43].

### 4.11. Melanoma

Myosin VIIA (Myo7a) is an unconventional myosin serving in intracellular movements. It was shown to promote proliferation, migration, invasion and tumour growth in B16 melanoma cells (a mouse model of skin cutaneous melanoma). Knocking down Myo7a decreases the expression of RhoGDI2 and Rac1 activity. Most interestingly, restoring the expression of RhoGDI2 in these cells is sufficient to also restore Rac1 activity, showing its direct implication in the process. Due to the involvement of Rac1 in multiple human cancers, it was inferred that Myo7a could exert tumour-promoting phenotypes by promoting RhoGDI2 expression levels and consequently Rac1 activity in B16 melanoma [47].

### 4.12. Lung Cancer

Knocking down RhoGDI2 significantly increased the proliferation, migration and invasion of A549 lung adenocarcinoma (LUAD) cells via increased Rac1 activity. Rac1 promotes migratory and invasive abilities in LUAD along with cytoskeleton rearrangements. When overexpressed, RhoGDI2 scatters F-actin filaments within the cytoplasm and induces loss of cell-cell adhesion and stress fibers formation. Contrary to what was observed in PDAC, overexpression of RhoGDI2 decreased the expression of Snail and increased the expression of E-cadherin, which might account for the anti-tumoural role of RhoGDI2 in LUAD which is suggested by correlation, showing an increased survival for patients having high tumour RhoGDI2 expression (Figure 5). Surprisingly, however, the reverse was observed for lung squamous cell carcinoma (Figure 5), again illustrating the critical importance of defining the different regulatory pathways involving RhoGDI2 in different cancer types.

As opposed to what was observed in HCC, downregulating the RhoGDI2 expression in LUAD increased the expression and activity of MMP9, which resulted in increased migratory and invasive abilities. Further studies showed that, in stark contrast to HCC, suppressing the expression of RhoGDI2 also increased the phosphorylation of Akt and PI3K, hinting that RhoGDI2 may exert anti-tumoural activities in LUAD by decreasing Rac1 activity and deactivating the PI3K/Akt pathway [48,49].

FHL1 (Four and a half LIM protein 1) is downregulated in multiple human cancers, including LUAD, where it exerts anti-tumoural activity [97]. As a potential underlying explanation, it was shown that FHL1 stimulates the expression of RhoGDI2, but not of RhoGDI1. However, the precise pathway is not yet elucidated.

## 5. Discussion and Future Directives

Although the role of RhoGDI2 in regulating the activity of Rho GTPases is well established, recent findings have highlighted complex molecular mechanisms of this regulation. In this review, we have presented the similarities and differences between RhoGDI1 and RhoGDI2 in regulating Rho GTPases and subsequently many biological processes. We also talked about the regulatory functions of RhoGDI2 in multiple biological processes such as immune response, HIV-1 replication and vascular remodeling and paid attention to its dual role in cancer.

It is important to note that despite sharing a high level of similarity in their structures, RhoGDI1 and RhoGDI2 have proven to be quite divergent in some functions. Both RhoGDIs may have different affinities for Rho GTPases or their co-regulators, which may cause them to regulate differently to Rho GTPases-dependent biological processes. In addition to their specific affinities towards Rho GTPases, the RhoGDIs may also be differentially distributed in the cytoplasm and/or membranes in different cell types, contributing to their diverging roles. However, the most crucial aspect to highlight is the completely divergent extreme N-terminal domains of RhoGDI 1 and 2. Both the RhoGDIs fold differently at their N-terminal domain (as highlighted in Figure 2B) and the N-terminal domain of RhoGDI2 is acidic in comparison to the N-terminal domain of RhoGDI1, thereby possessing a negative charge that may enable RhoGDI2 to establish long-range electrostatic interactions with co-regulators, whilst RhoGDI1 would be unable to do so.

As illustrated by their names, RhoGDIs were first considered only as inhibitors of the activity of Rho GTPases. However, as we have described, RhoGDI2 regulates many biological processes by not only inhibiting, but also promoting the activity of Rac1, as noticed in Jurkat T cell and myofibroblasts, and multiple human cancers [38,47,89]. To the best of our knowledge, RhoGDI1 does not promote the activity of key Rho GTPases. Further research should aim at explaining why RhoGDI2 is able to promote Rac1 activity in some cell types and not others by focusing on three main points: (i) the search for direct or indirect interactions between RhoGDI2 and third partners, such as the Rho GEFs Vav1 and Rac1-specific Trio [24,43,76]; (ii) investigation of the potential role of RhoGDI2 as a scaffold protein that can enhance the binding between Rac1 and actin-binding proteins, which would regulate its functions [42]; and (iii) analysis of dynamics of the subcellular localization of RhoGDI2, which can vary within different cell types, and also contribute to its dual role in regulating Rac1 activity.

As shown in Figure 6, RhoGDI2 is involved in several hallmarks of cancer, its expression correlates to either high or low aggressiveness in different human cancers. There are multiple drugs being developed to target other regulators and effectors of Rho GTPases [4,6,21]. Although no drugs specifically targeting RhoGDIs are in development, some drugs and traditional medicine involved in ameliorating cancer phenotypes have been shown to be implicated in different signaling pathways involving RhoGDI2. Therefore, inhibiting RhoGDI2 activity has therapeutic potential for some cancers and other diseases, but it is essential to tread carefully due to its complicated role in the human body. As an alternative, a better understanding of the implications of RhoGDI2 in pro- and anti-cancer pathways could lead to the identification of novel therapeutic targets. Further experiments aiming to decipher the intracellular molecular pathways affected by RhoGDI2 are required to appreciate its role in cancer biology.

A critical role of Rho GTPases, and therefore of RhoGDIs, is the organization of the actin cytoskeleton and all the downstream processes. Changes in the actin cytoskeleton affect other cellular responses such as changes in gene transcription, the microtubule cytoskeleton or vesicular transport. Besides transport, mobility and mechanical support, the microtubule cytoskeleton is also involved in chromosome segregation, centriole duplication and formation of primary cilia [98,99,100]. Rho GTPases have been extensively probed for their roles in processes involved in centriole duplication and cell division [101,102,103]. Despite the extensive research done to scrutinize RhoGDI2 for its involvement in multiple human cancers, no work has been done to investigate if and how RhoGDI2 is involved in the different steps regulating cell division.

RhoGDI2 has been shown to localize in centrosomes of human cervical cancer HeLa cells, wherein it is involved in centrosomal amplification, prolonged mitosis and aberrant cytokinesis. Mitosis is part of the cell cycle where the replicated chromosomes are separated into two new nuclei, and perturbations in the centrosomal function are a hallmark of cancer [104,105]. It will be interesting to investigate more functions of RhoGDI2 in the cell division cycle to increase our understanding of its involvement in cancer and other diseases to ensure appropriate targeting of RhoGDI2 for cancer therapeutics.

Besides its role in cancer cells, RhoGDI2 is highly expressed in immune cells where it regulates some of their key functions. Therefore, when considering studies correlating patient prognosis and RhoGDI2 expression, we must take into account the immune cells infiltrating the tumour microenvironment. This aspect of the role of RhoGDI2 in cancer progression has so far received little attention, while it could, however, have significant implications in cancer immunotherapy which is considered to be one of the most promising approaches for treating cancers.

## 6. Conclusions

Due to its dual role, it is crucial to study RhoGDI2 in the context of cancer. For this purpose, an enhanced characterization of its functions in cellular processes related to cancer, of its potential partners and of the dynamics of its subcellular localization are mandatory. However, one must remember to carry out these investigations not only in cancer cells but also in immune cells. Following this, RhoGDI2 can prove to be significant in cancer immunotherapy, which has made promising advances toward novel cancer therapeutics.

## Figures and Tables

**Figure 1 ijms-24-04015-f001:**
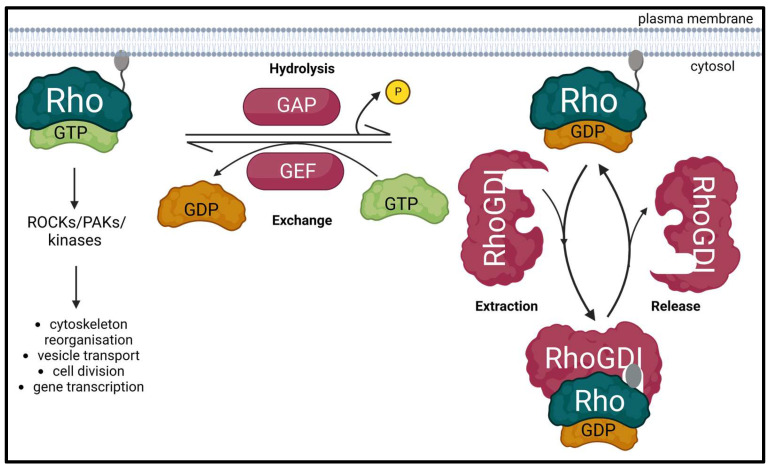
Schematic diagram of the Rho GTPase regulatory cycle. Inactive Rho GTPase is dissociated from its GDP and uptakes GTP to get activated through a process promoted by the Rho guanine exchange factors (RhoGEFs). Active GTP-bound Rho GTPase can then interact with its effectors such as Rho-associated coiled-coil containing kinases (ROCKs), PAK family of serine/threonine kinases (PAKs) and other kinases to participate in various biological processes. This interaction ceases when Rho GTPase-activating proteins (RhoGAPs) stimulate the hydrolysis of the bound GTP to GDP, thereby inactivating the Rho GTPase. The inactive GDP-bound form of Rho GTPase is free to bind to and be sequestered by Rho guanine nucleotide dissociation inhibitors (RhoGDIs). This induces their relocation to the cytoplasm, prevents their ubiquitin-mediated degradation and regulates the activation of Rho GTPases by GEFs. Created with BioRender.com (accessed on 28 January 2023).

**Figure 2 ijms-24-04015-f002:**
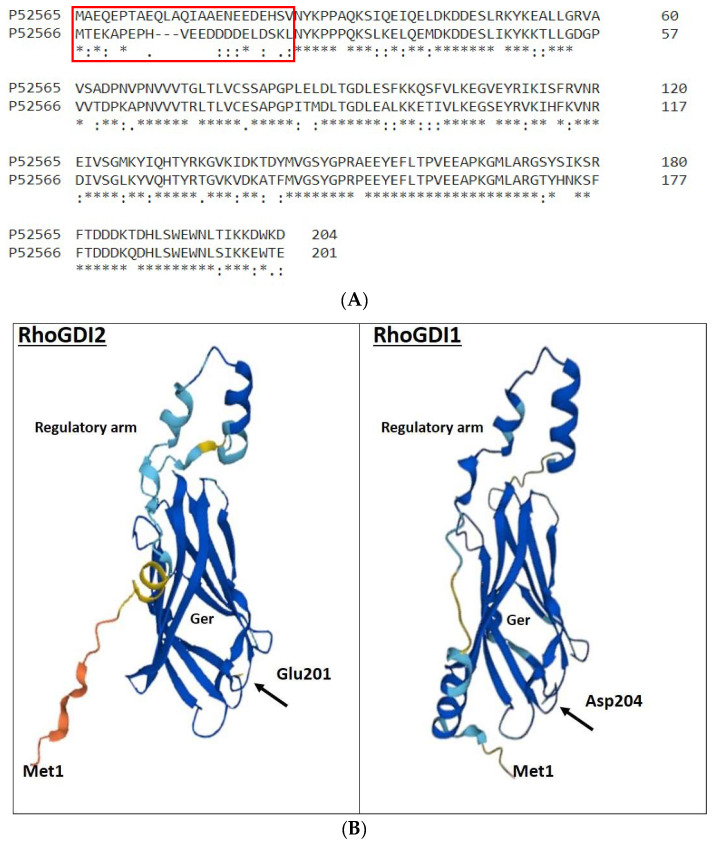
Comparison of primary and tertiary structures of human RhoGDI1 and RhoGDI2. (**A**) Protein sequences were compared using EMBL-EBI’s Clustal Omega tool. The accession numbers are as follows: human RhoGDI1, P52565 and human RhoGDI2, P52566. Identical residues are indicated by asterisks; substitutions for amino acids possessing highly similar or somehow similar characteristics are indicated by double and single dots, respectively. The highly divergent extreme N-terminal domains are enclosed in the red box. RhoGDI2 is phosphorylated at Y24, S31 and Y153 by β2 integrin-related kinases Src, c-Abl and Syk in response to PSGL-1 antibody ligation. On the contrary, RhoGDI1 is phosphorylated at S45, S48 and T52 by calcium-dependent protein kinase CPK3. (**B**) Predictions of the 3D structure of RhoGDI1 and RhoGDI2 are from the AlphaFold project (AF-P52565-F1 and AF-P52566-F1, respectively). Confidence regarding the 3D structure corresponding to different parts of the proteins is provided by color code, with dark blue representing the highest confidence and orange the lowest confidence. Ger: pocket accommodating the geranylgeranyl moiety of the Rho GTPases.

**Figure 3 ijms-24-04015-f003:**
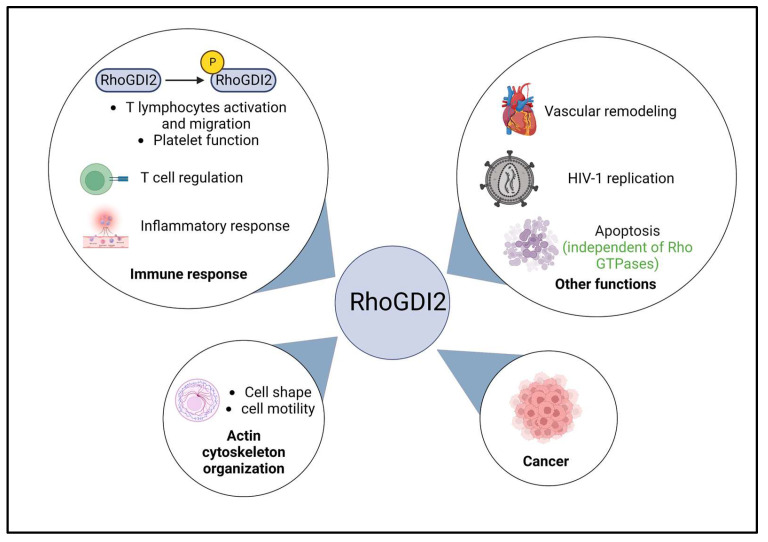
Schematic representation of the main functions of RhoGDI2 in the human body. RhoGDI2 is known to play a major role in multiple biological processes. It is involved in actin cytoskeleton organization, where it affects cell shape and movement. It is also involved in immune response wherein it is phosphorylated by kinases (Src, c-Abl and Syk) at Y24, S31 and Y153. Phosphorylated RhoGDI2 plays a role in T cell lymphocyte activation and migration and platelet function. RhoGDI2 is also involved in T cell regulation and inflammatory response. It regulates vascular remodeling by helping the migration of smooth muscle cells and is also implicated in HIV-1 replication and apoptosis. Lastly, RhoGDI2 is also involved in multiple cancers. Except for apoptosis, RhoGDI2 plays a Rho GTPase-dependent role in all other biological processes mentioned here. Created with BioRender.com (accessed on 28 January 2023).

**Figure 4 ijms-24-04015-f004:**
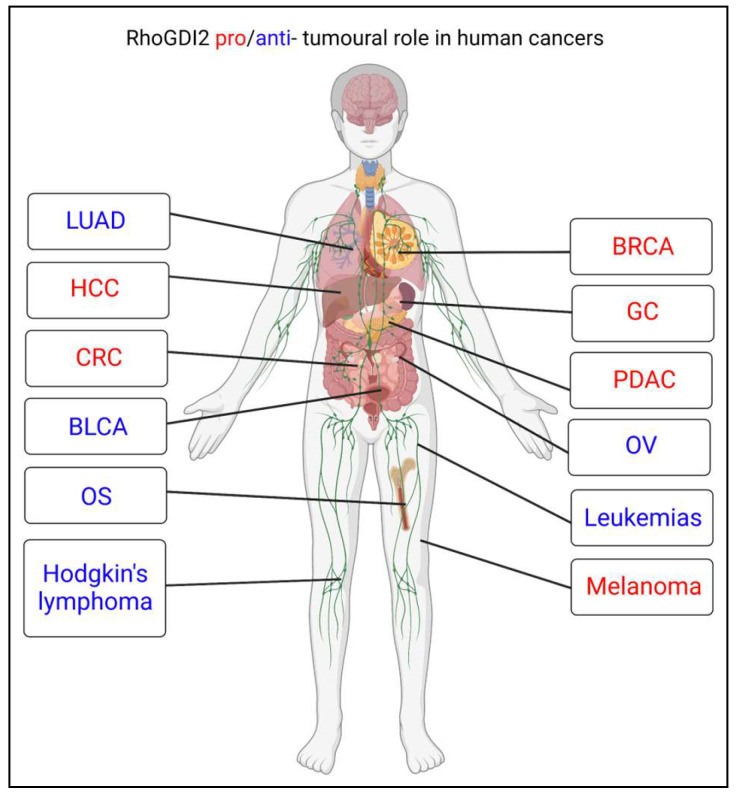
RhoGDI2 in cancer. Based on in vitro experiments and on correlation studies using human tumour samples, RhoGDI2 has been described as possessing either pro- or anti-tumour functions, depending on cancer types. It plays a tumour promoting role in hepatocellular carcinoma (HCC), colorectal carcinoma (CRC), breast cancer (BRCA), gastric cancer (GC), pancreatic ductal adenocarcinoma (PDAC) and skin cutaneous melanoma (SKCM). It plays a tumour suppressive role in lung adenocarcinoma (LUAD), bladder cancer (BLCA), osteosarcoma (OS), Hodgkin’s lymphoma, ovarian (OV) cancer and in multiple leukemias. Created with Biorender.com (accessed on 28 January 2023).

**Figure 5 ijms-24-04015-f005:**
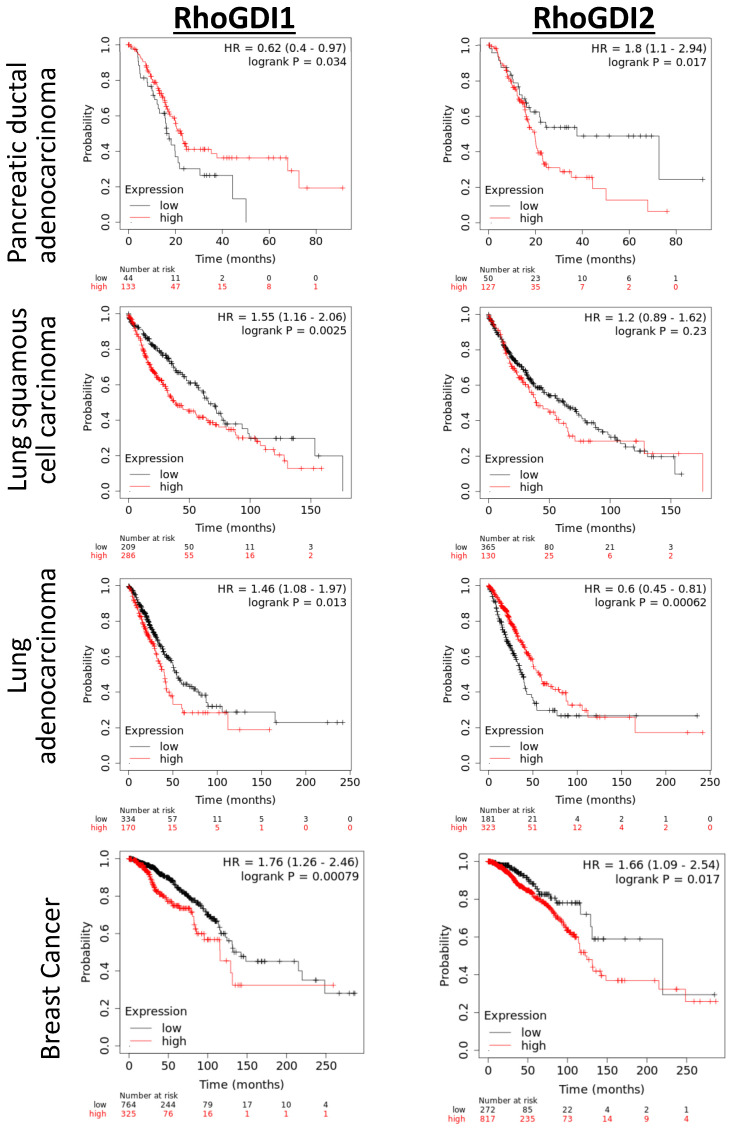
Correlation between patient survival and expression of RhoGDI1 and RhoGDI2 in few cancers. The Kaplan–Meier plotter (https://kmplot.com (accessed on 25 January 2023)) can be used to establish a correlation between the survival of patients and the level of expression of any gene in the primary tumour. Sources for the database include GEO, TCGA and EGA. The plots shown here were chosen to highlight the duality of RhoGDI2 expression in cancer that has either a good or poor prognosis, sometimes for cancer affecting the same organ (compare the curves for lung squamous cell carcinoma and lung adenocarcinoma). It also illustrates that the expression of RhoGDI2 and RhoGDI1 do not always correlate similarly to survival, as seen for pancreatic ductal adenocarcinoma, showing that they do not fulfill exactly the same function during cancer progression.

**Figure 6 ijms-24-04015-f006:**
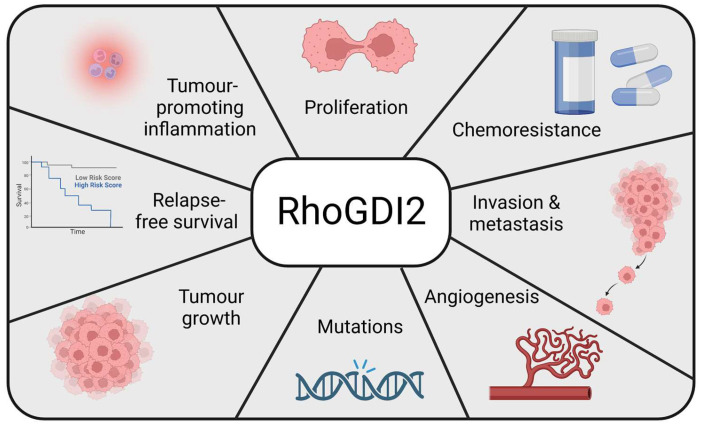
RhoGDI2 is involved in many hallmarks of cancer. RhoGDI2 is involved in many aggressive phenotypes of cancers via its regulation of Rho GTPases either directly or indirectly by interacting with other regulators of Rho GTPases. It modulates cell proliferation, tumour growth, cell migration, invasion and metastasis. It also provides chemoresistance and is shown to be mutated in some human cancers. It also contributes to tumour-promoting inflammation, angiogenesis and relapse-free survival in multiple human cancers. Created with BioRender.com (accessed on 28 January 2023).

**Table 1 ijms-24-04015-t001:** Expression of RhoGDI1 and RhoGDI2 in human cancers. Biphasic, up and then down.

Cancer	RhoGDI	Regulation	Reference(s)
Colorectal cancer	RhoGDI1	Up	[27]
	RhoGDI2	Up	[28]
Breast cancer	RhoGDI1	Up	[29]
	RhoGDI2	Up	[30,31]
		Biphasic	[32,33]
Hepatocellular carcinoma	RhoGDI1	Down	[34]
	RhoGDI2	Up	[35]
Bladder cancer	RhoGDI1	Down	[36]
	RhoGDI2	Down	[37]
Ovarian cancer	RhoGDI2	Down	[38,39]
Hodgkin’s lymphoma	RhoGDI2	Down	[40,41]
Gastric cancer	RhoGDI2	Up	[42,43]
Pancreatic cancer	RhoGDI2	Up	[44,45,46]
Melanoma	RhoGDI2	Up	[47]
Lung cancer	RhoGDI2	Down	[48,49]
Osteosarcoma	RhoGDI2	Down	[50,51]
Leukemias	RhoGDI2	Down	[52,53]

## Data Availability

The following publically available archives were analysed: www.uniprot.org; www.thebiogrid.org; and www.kmplot.com.

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
