# Peer review of "The Dual Function of RhoGDI2 in Immunity and Cancer"

_ijms, 2023, doi:10.3390/ijms24044015_

Round 1

Reviewer 1 Report

  • Review was nicely designed and presented with introduction, characteristics and regulatory functions of RhoGDI2 in general and going into specific detailing in cancer.
  • Each section is demonstrated with adequate details and the comparison between RhoGDI1 and RhoGDI2 is presented well.
  • Appropriate references were cited.
  • Abbreviations elaboration in beginning is recommended to remind the reader in the beginning of the article. 
  • Conclusion part can be more specific with additional detailing with respect to the future directives. 
  • Figure 1 can have each process labelled as it would be easy to follow.
  • In figure 1, explanation regarding Rocks/paks/kinases under the figure is recommended.
  • In figure 2A, font size increase is recommended.
  • In section 3.2.1, line 185 reference was mis-assigned/typo. It has to be [65] not [68].
  • In section 5. line 513, indent change is recommended to maintain consistency throughout the review.
  • doi's are missing for the references 11,31,32 and 81.
  • References should be consistent with respect to format of month and year. 

Author Response

Answers to Reviewer 1

  • Review was nicely designed and presented with introduction, characteristics and regulatory functions of RhoGDI2 in general and going into specific detailing in cancer.
  • Each section is demonstrated with adequate details and the comparison between RhoGDI1 and RhoGDI2 is presented well.
  • Appropriate references were cited.

Answer: We thank the reviewer for his/her positive comments on our manuscript.

  • Abbreviations elaboration in beginning is recommended to remind the reader in the beginning of the article. 

Answer: As recommended in the instructions for authors on the IJMS website, the abbreviations are defined the first time they appear. We have carefully verified in  the entire manuscript that all the abbreviations are defined at their first appearance.

  • Conclusion part can be more specific with additional detailing with respect to the future directives. 

Answer: We added precisions in the “Discussion and Future Directives sections”. A conclusion section has been also added at the end of the article.

  • Figure 1 can have each process labelled as it would be easy to follow.

Answer: Each process has been labelled in Figure 1.

  • In figure 1, explanation regarding Rocks/paks/kinases under the figure is recommended.

Answer: An explanation has been added in the legend of Figure 1.

  • In figure 2A, font size increase is recommended.

Answer: The font size has been increased in figure 2

  • In section 3.2.1, line 185 reference was mis-assigned/typo. It has to be [65] not [68].

Answer: The numbering of the references has been corrected.

  • In section 5. line 513, indent change is recommended to maintain consistency throughout the review.

Answer: The indent line 513 has been changed.

  • doi's are missing for the references 11,31,32 and 81.

Answer: The missing doi’s has been added except for two reference 11 and 81 for which doi’s are not available.

  • References should be consistent with respect to format of month and year. 

Answer: The format of the references has been corrected 

Reviewer 2 Report

The authors have displayed the significance of RhoGDI2 (guanine nucleotide dissociation inhibitor) in cancer. This review sheds 13 light on the dual opposite role of RhoGDI2 in cancer, highlights its underappreciated role in immunity, and proposes ways to explain its intricate regulatory functions. The review is well structured and has done an extensive study of the theme of the manuscript. The review is highly significant for researchers associated with cancer therapeutics. All the sections are well elaborated and correlated with each other. The concept is novel and highly relevant. However, there are some concerns that have to be resolved:

1. Title is not at all appropriate

2. Abstract section is not well structured and needs to be reframed.

3. Improve the quality of all figures. The figures are not clear and are a bit blurred.

4.  Conclusion section is missing.

Author Response

Answers to Reviewer 2

The authors have displayed the significance of RhoGDI2 (guanine nucleotide dissociation inhibitor) in cancer. This review sheds 13 light on the dual opposite role of RhoGDI2 in cancer, highlights its underappreciated role in immunity, and proposes ways to explain its intricate regulatory functions. The review is well structured and has done an extensive study of the theme of the manuscript. The review is highly significant for researchers associated with cancer therapeutics. All the sections are well elaborated and correlated with each other. The concept is novel and highly relevant. However, there are some concerns that have to be resolved:

Answer: We thank the reviewer for his/her positive comments on our manuscript.

  1. Title is not at all appropriate

Answer: The title has been changed to better fit with the content of our review.

  1. Abstract section is not well structured and needs to be reframed.

Answer: The abstract has been modified to be better structured.

  1. Improve the quality of all figures. The figures are not clear and are a bit blurred.

Answer: We have done our best to improve the quality of the figures. It seems that the resolution of the figures was decreased upon printing in PDF. We have modified the parameters to get better figures.

  1. Conclusion section is missing.

Answer: A conclusion section has been added.